# Intercomparison of global reanalysis precipitation for flood risk modelling

Fergus McClean[1], Richard Dawson[1], Chris Kilsby[1,2]

[1]School of Engineering, Newcastle University, Newcastle upon Tyne, NE1 7RU, UK

[2]Willis Research Network, 51 Lime St., London, EC3M 7DQ, UK

*Correspondence to*: Fergus McClean (fergus.mcclean@newcastle.ac.uk)

**Abstract.** Reanalysis datasets are increasingly used to drive flood models, especially for continental and global analysis, and in areas of data scarcity. However, the consequence of this for risk estimation has not been fully explored. We investigate the implications of four reanalysis products (ERA-5, CFSR, MERRA-2 and JRA-55) on simulations of historic flood events in

five basins in England. These results are compared to a benchmark national gauge-based product (CEH-GEAR1hr). The benchmark demonstrated better accuracy than reanalysis products when compared with observations of water depth and flood extent. All reanalysis products predicted fewer buildings would be inundated by the events than the national dataset. JRA-55 was the worst by a significant margin, underestimating by 40% compared with 14-18% for the other reanalysis products. CFSR estimated building inundation the most accurately, while ERA-5 demonstrated the lowest error in terms of

river stage (29.4%) and floodplain depth (28.6%). Accuracy varied geographically and no product performed best across all basins. Global reanalysis products provide a useful resource for flood modelling where no other data is available, but they should be used with caution due to the underestimation of impacts shown here. Until a more systematic international strategy for the collection of rainfall and flood impacts data ensures more complete global coverage for validation, multiple reanalysis products should be used concurrently to capture the range of uncertainties.

## 1 Introduction

The primary drivers of pluvial and fluvial flooding are precipitation events. The duration, intensity and spatial extent of these events can all affect the depth and extent of any flooding caused. Therefore, the choice of precipitation data when simulating floods is critical. Inaccurate precipitation will undoubtedly lead to a spurious and potentially misleading understanding of the risk posed by a given event. This effect is further exacerbated when low-quality precipitation data is used to project risk into

the future, with planning decisions being made based on the results. Unfortunately, understanding which source of precipitation is most appropriate is challenging. There is also spatial variation in the availability and quality of precipitation data. High-quality data is often collected by national or regional authorities but can be inaccessible or difficult to obtain, therefore continental or global precipitation datasets, such as reanalysis products, are a popular option despite their generally lower resolution and accuracy.


Reanalysis products are created by driving numerical models with recorded weather observations to build a comprehensive historical picture of a wide range of climatic variables. These datasets usually have global coverage and span multiple decades. Reanalysis precipitation data has been widely used in continental- and global-scale flood risk modelling (Winsemius et al., 2013; Alfieri et al., 2013; Andreadis et al., 2017; Pappenberger et al., 2012; Xu et al., 2016; Seyyedi et al.,

2015; Schumann et al., 2013). The main advantages of reanalysis products are their extensive spatiotemporal coverage and ease of access. In areas with a limited number of rain gauges that can provide high-quality observations, reanalysis products are often the best or only source of precipitation inputs for flood simulations. However, there is no guarantee that they are able to accurately represent extreme events and subsequently characterise flood risk. As a range of reanalysis datasets are available, there is also the question of which is more suitable for the application.


The influence of reanalysis data on flood risk estimates has previously been explored in part. Sampson et al. (2014) found that the loss ratio decreased by 8.5 times when using a reanalysis product (ERA-Interim) instead of a satellite rainfall product (CMORPH) in their catastrophe risk model of Dublin. Andreadis et al. (2017) compared flood models driven using an ensemble of parameters from 20CRv2 to a benchmark using observed flow boundary conditions and found that, overall,

20CRv2 only captured 15.7% of the benchmark inundated area. Mahto et al. (2019) used ERA-5, ERA-Interim, CFSR, JRA-55 and MERRA-2 to drive a macroscale hydrological model and simulate monsoon season runoff in India. CFSR and JRA-55 resulted in a strong positive bias, compared to a national precipitation dataset, while MERRA-2 strongly underestimated runoff. The two ERA products showed a much less prominent positive bias. While their study represents one of the first intercomparisons of different reanalysis precipitation products for runoff modelling, it does not go as far as looking at the

consequences for flood impacts. Meanwhile, Chawla et al. (2020) demonstrate a strong negative bias in flood discharge when using CFSR in the Himalayas. This is indicative of the spatial variability in accuracy inherent in reanalysis datasets, driven largely by assimilation data availability. Winsemius et al. (2013) compared flood impacts from the GLOFRIS model cascade, which uses ERA-Interim, with The OFDA/CRED International Disaster Database (EM-DAT). River flood risk maps and damage estimates produced using ERA40 and ERA-Interim were found to be in the same order of magnitude as

estimates from EM-DAT and the World Bank. However, the effect of using a different source of precipitation was not assessed and therefore the impact of using reanalysis data on the cascade is unknown.

This paper extends previous studies by undertaking a systematic intercomparison of how modern reanalysis products compare when used to drive a hydrodynamic flood model. This provides important insights to inform the selection of data

for flood modelling in data-sparse regions as well as a more general assessment of how well extreme rainfall events are captured in each product. To provide further context and identify the potential effects on flood risk assessments, the flood model outputs are subsequently used to estimate the number of buildings that would be inundated by each rainfall product. While this study provides an example of how varied results may be between products, the relative performance of each dataset may differ between areas and events and is not necessarily transferable.

## 2 Methodology

### 2.1 Study Area

To assess the performance of global reanalysis precipitation, more reliable gauge-based data is required as a baseline to validate against. However, the quantity and quality of gauge observations are limited across much of the globe, particularly in sparsely populated and poorer regions. Local gauge data may in fact be of lower accuracy than the large scale products if the rain gauges on the ground are of poor quality or have been influenced by human error. There is no way to check which is more correct by looking at precipitation alone and an independent source of data is required. River flow data has been used for this purpose in the past (Beck et al., 2017) and presents a viable option for assessing precipitation performance in the context of flood events. To fulfil the requirements of high-quality local precipitation and river flow observations, an area of northern England, encompassing the Tyne, Tees, Eden, Wear and Lune basins (Figure 1) was selected for this study. The relatively simple flood response of these steep, surface water dominated basins, and the occurrence of recent flood events, means they provide a suitable testbed for investigating the effects of using global reanalysis products for more localised flood risk modelling.

### 2.2 Model Setup

The City Catchment Analysis Tool (CityCAT) (Glenis et al., 2018), a hydrodynamic surface water flood model, was used to simulate flooding in this study. CityCAT represents spatial rainfall fields falling directly onto elevation surfaces made up of uniformly sized square grid cells and propagating according to the shallow water equations (SWEs). The model uses the method of finite volumes and shock capturing schemes to solve the SWEs with a Generalised Osher-Solomon Riemann solver. The system is suitable for this study as it is able to directly capture the effects of rainfall on flood depths without requiring any intermediate steps. The domain grid of each simulation is directly generated from the DEM and has the same resolution, in this case 50m. Model domains within the study area were delineated using HydroBASINS (Lehner and Grill, 2014). The outer boundaries of the domain were treated as being open which allows water to exit the domain at basin outlets. No processing of the DEM was undertaken and it was used in its original form.

The DEM does not explicitly include river bathymetry, except where the river is of similar or greater width than the DEM resolution, and within low points in valleys. For example, in Carlisle where we undertake more detailed impacts analysis this includes a coarse rectangular channel bathymetry, with depths ranging from 1.5-4m and a width of 50-100m. (2021)(2019) Although Neal et al (2012) showed that representation of channels is important for accurate simulation of flood propagation, Neal et al (2021) and Dey et al (2019) show that choices about how bathymetry is represented becomes less important at more extreme return periods. Peña et al. (2021) go as far as to conclude that: "small-scale features and river bathymetry are negligible under extreme hydrologic events as the floodplain conveyance capacity is the driving principle of flood inundation dynamic". By not explicitly embedding an accurate river bathymetry into our model , it is likely that flood

extent will be over-estimated and channel discharge underestimated, although the model performance appears adequate in this regard. Whilst the lack of an explicit river channel should be considered when interpreting absolute measures of accuracy, it is a reasonable approximation here as we are not studying bathymetry or DEMs, we focus on large flood events, and any errors apply to all rainfall simulations allowing for objective intercomparison of global rainfall products which is the purpose of this study.

Water depths were output every hour for each grid cell within the domain and then extracted at each gauge location (Figure 1). The Manning's coefficient for all domains was uniformly defined as 0.03 (Chow, 1959), that is the same, or similar, to other studies (Choné et al., 2021; Addy and Wilkinson, 2021; Hou et al., 2020). The land surface was assumed to be impermeable given the extreme nature of the selected events (described below). Once the ground is saturated during long-duration flood events, subsurface processes will cease to have a large impact on water levels, especially for catchments such as those addressed here with generally shallow soils and low base flow indices. For example, on a small catchment Hossain Anni (2020) found an increase of only 0.02m in average flood depth when excluding infiltration from a 100-year flood model. Furthermore, larger scale studies by Ni et al. (2020) and Hou et al. (2021) show that peak flow and flood extent are relatively insensitive to infiltration rates, although an assumption of no infiltration would impact outflows as the flood wave falls; the effect of this is greater for longer floods, and would be more significant in semi-arid or arid regions (which is not the case here). In the case of the Carlisle flood in the Eden basin it is documented that antecedent conditions had led to saturated soils when the flood event occurred (Convery and Bailey, 2008). Additionally, it is assumed that there are no artificial water abstraction measures or flood defences present, information on the elevation of the latter in 2005 could not be located. This configuration of friction, bathymetry and infiltration parameters is sufficient as the primary aim of this study is to compare the influence of different global rainfall products rather than the absolute accuracy of the hydrodynamic model.

## 2.3 Rainfall

Four global reanalysis products (JRA-55, MERRA-2, ERA-5 and CFSR) have been selected and compared against CEH-GEAR1hr, used here as a benchmark. Each rainfall dataset is described below and key characteristics are shown in Table 1. The reanalysis products were selected based on their high spatiotemporal resolution, open availability and suitable duration. Events between the start and end dates of CEH-GEAR1hr (1990-2014) were selected based on the peak stage at the most downstream river gauge within each basin (Table 2). This identified the most extreme rainfall events, independently of the rainfall data itself. The largest events were chosen as they have the greatest impact in terms of flood damages. Looking at a wider range of events may have provided a more comprehensive view of the performance of reanalysis products across different magnitudes, however this was outside the scope of the study. Different events were selected for each basin as the largest extremes may have occurred at different times in different areas. Each identified event was only simulated in the basin in which it was observed, to enable river gauge records to be used for validation. Simulations were commenced two

weeks before the discharge peaks and ran until one week after. This was to allow model spinup and characterisation of hydrograph recession. The sensitivity to run duration was not explicitly assessed here but the duration was sufficient in all cases to ensure adequate accounting for antecedent rainfall and return to normal flow conditions. Antecedent rainfall is necessary to initiate normal flow in the river channels, which requires the water from all upstream cells to reach the outlet of the basin. Normal flow here refers to the flow in the channel before the flood event took place. If no spin-up period is included, then flood magnitudes would be underestimated, and the flood wave would not propagate in a physically realistic way.

The events, according to each dataset, are mapped in Figure 3 and compared with time series of observations at selected gauges in Figure 2. CEH-GEAR1hr contained, on average, higher rainfall totals than the reanalysis products and JRA-55 represented only approximately half as much precipitation as other reanalysis products. ERA-5 significantly over-estimated the gauged precipitation for the 2005 event in the Tyne basin but other than this, reanalysis products underestimated rainfall totals. The 1995 event in the Lune basin was the most under-represented across the reanalysis products. CEH-GEAR1hr was consistently very close to the gauge observations as they will have been used as part of its creation.

Each rainfall value was converted into a rate (kg m-2 s-1) at the corresponding times and each point on the original reanalysis grid was converted into an area with a width and height equivalent to the horizontal and vertical resolution of the dataset. This resulted in differently sized rainfall polygons for each dataset, corresponding to the resolutions listed in Table 1. These areas were then re-projected into British National Grid as cartesian coordinates are required by CityCAT. The rainfall products are described below, along with findings from previous studies which have assessed their performance.

The Centre for Ecology and Hydrology provide an hourly version of their Gridded Estimates of Areal Rainfall dataset (CEH-GEAR1hr) (Lewis et al., 2018). This hourly product is based on a daily product which interpolates data from rain gauges using natural neighbour interpolation (Tanguy et al., 2019). CEH-GEAR1hr uses nearest neighbour interpolation to maintain more realistic weather patterns and unmoderated peak values. To ensure consistency between the hourly and daily versions, the daily totals were maintained in the hourly dataset by scaling the interpolated values accordingly (Lewis et al., 2018). Quality control procedures were applied to the hourly gauge data used to produce the gridded product. Each gauge was compared with CEH-GEAR daily and 92.9% matched well. Other flags were applied to suspicious values, such as those which exceeded the 1- or 24-hour record values, values which were preceded by 23 hours of no rain and for tipping bucket gauges where the frequency of tips was unexpectedly high. Combinations of these flags were used to identify and exclude values where necessary. This gauge-based dataset was used as a baseline to compare against the reanalysis products identified below. The approach of using a gauge-based product as a baseline is well established (Jiang et al., 2021; Sun and Barros, 2010; Lei et al., 2021). However, there are always errors present in any rainfall product and gauge-based products are no exception. For example, wind-induced under-catch may lead to a negative bias in precipitation (Pollock et al., 2018).

The network may also not be dense enough to capture given events effectively. Despite these limitations, CEH-GEAR1hr provides the best available gridded hourly data for the UK, based on quality-controlled data from a well-established network.

Japanese Meteorological Agency reanalysis 55 (JRA-55) replaces JRA-25, incorporating higher resolution and better data assimilation, among other improvements (Kobayashi et al., 2015; Japan Meteorological Agency, Japan, 2013). Suzuki et al. (2018) were able to effectively simulate continental river discharge using JRA-55, however, they found large biases attributable to precipitation error in some regions. Chen et al (2014) found that JRA-55 has a diurnal difference of ~50% which was comparable to an observed satellite dataset from the Tropical Rainfall Monitoring Mission (TRMM). Hua et al (2019) found that JRA-55 over-estimated rainfall by ~15% around southern Sahel and western equatorial Africa. Over eastern China, Chen et al (2019) found that rainfall was better represented by JRA-55 than ERA-Interim, CFSR and MERRA-1. Meanwhile in the Northern Great Plains, JRA-55 has been found to perform worse than other reanalysis products such as ERA-5 and MERRA-2, demonstrating a strong wet bias (Xu et al., 2019). The variability in performance across these studies illustrates that the accuracy of JRA-55 is not consistent between regions.

Modern-Era Retrospective Analysis for Research and Applications 2 (MERRA-2) (Global Modeling and Assimilation Office, 2015) builds upon its predecessor, MERRA (Rienecker et al., 2011) with reduced biases in aspects of the water cycle, among other improvements (Gelaro et al., 2017). MERRA-2 uses observed precipitation products to correct the forecasts and provide better estimates (Reichle et al., 2017). Hua et al. (2019) found that MERRA-2 was better at representing rainfall climatology over Central Equatorial Africa than ERA-Interim and JRA-55, among others, with a mean bias of only 0.01 mm/day. Hamal et al. (2020) found that MERRA-2 was able to accurately capture the seasonal precipitation in Nepal when compared to gauge observations (R ≥ 0.95). Over Pakistan, MERRA-2 precipitation has been shown to have an RMSE of 1.68 mm and performed better than JRA-55 (2.2 mm) but not as well as ERA-5 (1.53 mm) (Arshad et al., 2021). Liu et al. (2021) found that MERRA-2 precipitation was more similar to satellite observations during summer in the Sichuan Basin, with a linear correlation coefficient of 0.9 compared to other parts of the year with 0.57. In India, MERRA-2 has negative bias of 10% in extreme rainfall, compared with 33% from ERA-Interim (Mahto and Mishra, 2019).

The European Centre for Medium-Range Weather Forecasts Reanalysis 5 product (ERA-5) (Hersbach et al., 2020) replaces and improves on ERA-Interim (Dee et al., 2011), which stopped being produced in August 2019. It supports an increased spatial and temporal resolution, along with an updated modelling and data assimilation system, which has resulted in better representation of convective rainfall (3.8% vs -5% median bias in monsoon precipitation over India)(Mahto and Mishra, 2019). The land surface component is being used to calculate river discharge for the Global Flood Awareness System (Harrigan et al., 2020). Albergel et al. (2018) found that ERA-5 resulted in better estimates of river discharge than ERA-Interim when used to drive a land surface model of the US. It has also been shown to outperform a range of other reanalysis products as part of a hydrological model applied in two Indian basins, with an RMSE of 25.5% compared with 40.2%,

59.2% and 75.6% from CFSR, JRA-55 and MERRA-2 respectively (Mahto and Mishra, 2019). In Pakistan, ERA-5 has an RMSE of 1.53 mm compared with daily gauge data, again outperforming MERRA-2 and JRA-55.

The NCEP Climate Forecast System Reanalysis (CFSR) (Saha et al., 2010a) replaces the previous NCEP/NCAR reanalysis (Kalnay et al., 1996) and uses a very similar analysis system to MERRA-2 (Saha et al., 2010b). Zhu et al. (2016) demonstrated that CFSR was liable to overestimate high streamflow in two Chinese basins using SWAT and highlighted that performance varied between basins (19.15% - 31.47% bias). Nkiaka et al. (2017) found that using CFSR over ERA-Interim

resulted in substantially improved representation of river flow in the Sudano-Sahel Region, with maximum Nash Sutcliffe Efficiencies of 0.43 and -0.56 respectively. In the Amazon basin, CFSR has been shown to underpredict winter precipitation with a bias of -0.60 and overpredict summer precipitation with a bias of 0.11 (Blacutt et al., 2015). During 2010-2014 in Bangladesh, CFSR overestimated precipitation relative to gauge observations with a bias of 1.18, this was greater than ERA-5 which had a bias of 0.80 (Islam and Cartwright, 2020). In the Johor River Basin, Malaysia, the daily RMSE of CFSR was

found to be 17.70 mm when compared with gauge observations (Tan et al., 2017).

## 2.4 Digital Elevation Model

The terrain dataset used to represent the domain surface is a nationally and freely available Digital Elevation Model (DEM) product from the Ordnance Survey, known as OS Terrain 50 (OST50) (Ordnance Survey, 2017). This has been shown to perform better than a range of global DEMs for flood risk modelling (McClean et al., 2020). The product is based on a

215 combination of photogrammetry and topographical surveys and is corrected using a combination of automated and manual processes to create a bare earth surface with raised structures removed. The DEM was clipped to the area of each basin and used directly within the models (Figure 1).

Other, higher resolution DEMs are available, such as Environment Agency LiDAR, which h typically have considerable

advantages for flood modelling (Sanders, 2007; Muhadi et al., 2020; Md Ali et al., 2015; Trepekli et al., 2022). However, complete LiDAR coverage of the study area was not available. The DEM is therefore a source of uncertainty and is likely to cause over-estimation of inundation. For example, Yunus et al. (2016) found that using OST50 resulted in 3-10% more inundation for London when compared to 1m LiDAR. Even if LiDAR was available for the entire domain of each basin included here, the data would not be able to be used at its original 1-2 m resolution and would have to be resampled to a

lower resolution to enable the simulations to complete in a reasonable time period. An alternative approach might be to merge LiDAR where available with OST50. However, either of these approaches would reduce the benefits of using LIDAR over OST50 and the choice of resampling method would introduce a new uncertainty as it also influences model outputs..

**2.5 Validation Data**

15-minute stage observations were obtained from the Environment Agency (EA) at the most downstream gauge in each of the five basins via a Freedom of Information request. The most downstream gauge was used in each case as these are influenced by the largest areas of rainfall. The IDs, catchment areas and locations of each gauge are listed in Table 2. Observations of flood extent during the event in Carlisle in the Eden basin were extracted from the EA Recorded Flood Outlines dataset (Environment Agency, 2019a). Distributed measurements of maximum water depth from the same event were provided by Neal et al (2009).

**2.6 Exposure**

Building outlines from OS VectorMapLocal (VML) (Ordnance Survey, 2018) were used to estimate numbers of buildings inundated by each model. VML only represents individual buildings with a floor area over $20\,\mathrm{m}^2$ and each polygon may represent multiple buildings. Therefore, not all buildings are included in the inundation totals. This is acceptable for this analysis which compares the relative magnitude of flooding, rather than the absolute totals. Buildings from VML were classified as flooded if they intersected any model cell above a typical property threshold of 0.3 m (Environment Agency, 2019b).

**3 Results**

The performance of each simulation was compared in terms of the magnitude and timing of the hydrograph peak, the flood depth and extent, and the number of buildings inundated (Table 3). ERA-5 outperformed other reanalysis datasets in terms of hydrograph peak error and floodplain depth, however, CFSR produced more similar inundation levels to CEH-GEAR1hr and demonstrated more accurate peak timing, flood extent and depth compared to point observations. JRA-55 performed significantly worse than other reanalysis products across all measures. The variability of each metric will now be assessed in more detail, including spatial variations in performance.

The maximum water depths according to models using each of the rainfall datasets are shown in Figure 4. Overall, the spatial distribution of floodwater is similar, as the same DEM is used in all models. There are noticeably higher depths along main river channels in the CEH-GEAR1hr results. JRA-55 presents less clearly visible channels than the other models, particularly in the Lune and Tees basins. The maps also illustrate that the MERRA-2 model produced lower depths in the Tyne basin than other reanalysis precipitation datasets. Across all basins, ERA-5 and CFSR produced similar distributions of error relative to the CEH-GEAR1hr results. The inter-quartile range of errors in MERRA-2 is narrower but the median error is slightly further below zero than ERA-5 and CFSR. JRA-55 water depths were significantly further below the other reanalysis datasets.

The flood extents from each model during the 2005 event in the Eden basin are compared against Environment Agency recorded flood outlines in Figure 5. This event was chosen because it provided the largest available flood extents in the observed dataset which coincide with a built-up area. CEH-GEAR1hr resulted in the highest Critical Success Index (CSI) (0.54). This level of performance with rainfall inputs derived from rain gauge data is consistent with the findings of Bárdossy et al. (2022) which showed that up to 50% of model error can be attributed to precipitation uncertainty. CFSR and ERA5 performed similarly to each other and less well than CEH-GEAR1hr. MERRA-2 caused further underestimation of extent, while JRA-55 had the lowest CSI by a significant margin. The recorded outlines data does not contain information about the total area that was surveyed so regions incorrectly identified as flooded in the model outputs may still have been flooded in reality. This is clearly the case in the downstream section of the Eden in the upper left of each plot, along with other water courses visible in the model outputs. This means that the CSI values are under-representative of accuracy, however they provide a useful metric for comparison between datasets.

Wrack and water marks recorded following the 2005 event in Carlisle in the Eden catchment (Neal et al., 2009) have been compared against maximum modelled water depths in Figure 6. Wrack marks are left by debris deposited at the flood edge (HR Wallingford, 2004), while water marks are left as stains on the side of structures within the flooded area. The DEM values from the computational grid were subtracted from the observed water elevations to produce flood depths for comparison with the model outputs. Any depths which were calculated as being below zero were assumed to be zero. Again, CEH-GEAR1hr is the closest to the observed data with an RMSE of 0.41m, followed by CFSR and ERA-5 with approximately twice the error. JRA-55 resulted in less than half the $r^2$ value of other datasets. The ranking of datasets remained consistent between the CSI analysis and comparison against observed depths. Without data on the 2005 flood defence crest levels it was not possible to incorporate them. However, the storm was estimated to be a 1 in 170 year event (Environment Agency & Cumbria County Council, 2016) (Environment Agency and Cumbria County Council, 2016), far higher than the design return period for many fluvial flood defences, so whether by overtopping or floodplain flow they would be expected to have a relatively minor influence on this event.

Time series of water depths were extracted from the models at each river gauge location and compared with the observed values (Figure 7). In all basins, apart from the Wear, CEH-GEAR1hr was closest to the observed peaks and predicted the highest maximum depth. In the Wear basin, where all models overestimated river stage, CEH-GEAR1hr was actually the least accurate. However, the observed values may be misleading here as flows go out of bank above 3 metres and so peaks are truncated in the observed series. This means that the actual stage is likely to be higher than the recorded values and therefore the magnitude of the reported errors overestimated. JRA-55 consistently severely underestimated river stage and only captured peaks in the Eden and Tyne basins. ERA-5 and CFSR display relatively similar performance across all basins. Meanwhile, MERRA-2 underestimated the peaks in the Eden and Tyne. All reanalysis products strongly underestimated the flood peak in the Lune basin.

The total numbers of inundated buildings for each model are shown in Figure 8. As there is no observed building inundation data available, it cannot be concluded which is the most accurate. However, a reasonable assumption might be that since CEH-GEAR1hr is based on rainfall observations and is at a higher resolution, it is likely to produce the closest estimate to the truth. In four out of five basins, using CEH-GEAR1hr resulted in the highest number of inundated buildings. ERA-5 inundated the most buildings in the Tees basin despite not being consistently higher than the other reanalysis datasets in the other basins. JRA-55 inundated the lowest number of buildings by a large margin in all basins apart from the Tyne, where it exceeded both MERRA-2 and CFSR. CFSR never resulted in either the highest or lowest number of inundated buildings. There is general agreement between the rankings of modelled peak water depth as shown in Figure 7 and the number of inundated buildings. Notable exceptions include the Tees, where the high inundation levels predicted by ERA-5 were not replicated in its depth peak, which was lower than CEH-GEAR1hr by a clear margin. Changing the inundation threshold had only very minor effects on the relative differences in inundation between datasets.

## 4 Discussion

The underestimation of extreme rainfall by reanalysis products has previously been identified in the literature (Blacutt et al., 2015; He et al., 2019; de Leeuw et al., 2015). The results presented above demonstrate that this leads to a persistent bias towards underestimation of flood depths and impacts when using global reanalysis products in place of high-resolution gauge-based rainfall datasets. One contributing factor is that the model grid resolution of the global climate models (GCMs) used may not be high enough to capture the dynamics of extreme events. Seasonal and local characteristics may also not be captured by the GCMs. Any resulting negative bias in precipitation propagates into flood depths and impacts as less water enters the hydrodynamic model and accumulates on the floodplain. The negative bias has been shown to exist in depths across the basins studied, at river gauging stations and specifically at the locations of buildings, which correspond to built-up areas exposed to flooding. This finding is in line with Sampson et al. (2014), who show ERA-Interim, an older product, underestimated flood risk. Our results, however, do not indicate such a stark bias, perhaps because the products used here are more modern and advanced than ERA-Interim. This is backed up by Towner et al. (2019) who have demonstrated improved performance of ERA-5 over ERA-Interim using hydrological models of the Amazon. Hirpa et al. (2016) also illustrate that ERA-Interim can underestimate flood risk, with spatial variability, which further reinforces our finding. In contrast, Andreadis et al. (2017) find flood extent to be overestimated (relative to a benchmark simulation) when using the 20CRv2 reanalysis product. However, they did find that outflow discharge was underestimated, which agrees with our results. Their assessment of flood extent did not include flood depths or effects on the inundation of exposed assets, as we have done here, which may explain the observed overestimation to some degree. We also did not replicate the underestimation of streamflow found by Zhu et al. (2016) when using CFSR. Though, it is difficult to draw direct comparisons given the major differences in methodology between studies.

We found that no precipitation product performed better in all models and each product performed differently depending on the basin. This implies that the optimum dataset to use depends on the location of the model. JRA-55 was very poor at capturing extreme rainfall and subsequently hydrograph peak and inundation magnitude in almost all cases. This may be slightly influenced by the lower temporal resolution, but it is unlikely that the small difference in observation frequency would result in such a strong negative effect on model performance. ERA-5 consistently performed better than other reanalysis datasets in terms of capturing the observed hydrograph peak, apart from in the Eden, where CFSR was more accurate and also demonstrated a higher CSI and lower RMSE relative to recorded outlines and wrack mark depths. ERA-5, CFSR and MERRA2 were more evenly matched in terms of floodplain water depth (Figure 4), and impacts (Figure 8). We find no cause to favour any of these three datasets and suggest that all three could be adopted in parallel by reanalysis-based flood models to capture the range of uncertainty.

Links between hydrograph performance and estimated numbers of inundated buildings are present but the relationship is not consistent. For example, in the Tyne basin, CFSR estimates a higher gauge peak than JRA-55 but, at the same time, inundates fewer buildings. Meanwhile, MERRA-2 only has the lowest hydrograph peak in the Tyne, where it estimates the lowest total building inundation compared to other models. CEH-GEAR1hr is also both generally higher in terms of both building inundation and hydrograph peak, but the occasions where this is not the case do not correspond to the same basin. These findings demonstrate that there is generally a positive relationship between peak hydrograph depth and numbers of inundated buildings, but increased river depth does not always lead to greater inundation. Therefore, hydrograph performance is not an entirely reliable metric for assessing the accuracy of flood risk estimated using global reanalysis products.

The physically-based 2D hydrodynamic model achieves a good fit to several indicators, despite a reported Root Mean Squared Error (RMSE) for the OST50 DEM of 4m. A key reason for this is that the RMSE is the absolute accuracy of elevation across the whole country. This combines systematic (e.g. block linkages between photogrammetric observations) and random errors from one end of the country to the other, and is therefore not an absolute measure of accuracy for a given area of interest. Local precision, the relative accuracy from point to point, is more important here and will be much better than the RMSE over the (relatively) small river basins (e.g. RICS (2021) suggest an Area of Interest would likely have relative precision of the order of decimetres). Moreover, OST50 has been validated to meet the positional requirements for key features such as waterbodies, and to capture topography (Ordnance Survey, 2022). Although Yunus (2016) showed it likely led to a small overprediction of flood impacts relative to LiDAR, it has been used successful for hydrological modelling (Chen et al., 2021). The combination of relative accuracy and positional validation against key features of the DEM explains the performance of the hydrodynamic model. Whilst the hydrodynamic performance was reassuring, calibration could have further improved model fit (Maggioni and Massari, 2018). However, the focus of the study is on the

sensitivity of model performance to different global rainfall products. We have, by choice, therefore not adjusted any input
data, or undertaken any specific calibration as this would compensate, differently for each catchment being studies, for the
errors and differences in the data that this work is seeking to understand.

The underestimation of inundation magnitude caused by using global precipitation data is counter to the overestimation that
results from using global DEM data, as demonstrated by McClean et al. (2020). The negative inundation bias caused by
using reanalysis precipitation is, however, not as strong as the positive bias from global DEM products. This is because
changes in rainfall input have a less significant impact on the spatial distribution of flooding than changes in DEM input.
Therefore, it is anticipated that the combined effects of using both global DEM and global reanalysis precipitation would not
cancel themselves out and are likely to produce a net positive bias.

Undoubtedly the effects shown here are specific to the study area and other locations may present different patterns. Each
reanalysis product may behave differently across climatic regions, for example. Similarly, the assumption of no infiltration
would lead to increased underestimation of flows in arid climatic regions. Areas with highly constrained topography are
unlikely to be strongly affected by the choice of precipitation data, in terms of flood extent and numbers of inundated assets
because increases in total rainfall volume will not greatly alter flood extent if there are no new available flow pathways.

A key limitation to applying our methodology in other locations is the requirement for high-quality gauge-based
precipitation datasets and river stage observations to compare against. Despite the caveat of locality, our results do
demonstrate the potential for underestimation of flood risk when reanalysis products are involved. This underestimation has
been observed in other areas using earlier reanalysis rainfall products (Sampson et al., 2014) and users of models based on
reanalysis data should be aware of this effect.

## 5 Conclusions

Using precipitation from global reanalysis datasets results in an underestimation of flood risk by 14-18 % of inundated
buildings (Table 3, excluding JRA-55 as it was far outside the range of other products) relative to CEH-GEAR1hr, which
outperformed reanalysis products in terms of flood depth and extent when compared to observations. The effect is location-
specific, though, and this study found that no product performed best across all five of the catchments we studied. In some
areas, the reanalysis data did result in similar levels of inundation to the national observed precipitation product. This is a
positive message for the use of reanalysis data in flood risk modelling generally and future progress in forecast models will
undoubtedly reduce this gap even further.

As climatic and land-use changes increase flood hazard, the importance of accurately understanding current and future flood risk is increasing. Reanalysis data has enabled flood risk assessments to be undertaken more widely. However, this analysis shows global or regional reanalysis data should not yet be considered as a replacement for local, high resolution, observations. Uncertainties in flood risk assessment using reanalysis data need to be properly quantified and communicated to insurers, local and national authorities and communities, to ensure flood risk management decisions are not misinformed.


While reanalysis datasets do show promising and improving results (ERA-5 achieved a mean absolute hydrograph peak error of 29.4 %, equivalent to CEH-GEAR1hr and CFSR only inundated on average 14.4 % fewer buildings than CEH-GEAR1hr), caution should be used when interpreting outputs from any models based on them due to the underestimation of inundated buildings demonstrated here. However, as no observed building inundation data were available, our findings are

not definitive. We suggest that multiple products, such as ERA-5, CFSR and MERRA-2, should be used where possible to capture the full range of rainfall uncertainty. This is because each of these products has been shown to perform better in different areas or when using different performance measures. Based on the comparatively strong negative bias in inundation and flood peak shown here for a limited set of events, JRA-55 may result in substantially lower risk estimates than other reanalysis products and users of model outputs based on it should take this into account. However, as highlighted, certain

products may perform better in other areas and further research is needed to assess new and existing reanalysis products for flood modelling across a wider range of climatic regions. To enable this, a more systematic international strategy for the collection of rainfall data is needed to ensure more complete global coverage of validation data, building on efforts from Lewis et al. (2019). New reanalysis products continue to be developed which may improve on the findings presented here (Muñoz-Sabater et al., 2021) while bearing in mind the findings of Bárdossy et al. (2022) on the fundamental

uncertainty of the reference raingauge data. This will require ongoing validation efforts to identify possible advancements in terms of flood risk analysis

**Code/Data availability**

The reanalysis products can be downloaded using the DOIs in Table 1. All model results and code used to generate figures can be accessed at https://doi.org/10.25405/data.ncl.21681101.

**Competing interests**

The authors declare that they have no conflict of interest.

**Author contribution**

FM processed the data and executed the simulations. FM prepared the manuscript with contributions from all co-authors.

**Acknowledgements**

The authors were supported by the Natural Environmental Research Council (NERC) Centre for Doctoral Training for Data, Risk and Environmental Analytical Methods (DREAM), Engineering & Physical Science Research Council TWENTY 65: Tailored Water Solutions for Positive Impact (EP/N010124/1), UK Research & Innovation Global Challenges Research Fund Water Security and Sustainable Development Hub (ES/S008179/1), and Willis Research Network. The City Catchment Analysis Tool (CityCAT) was developed and provided by Vassilis Glenis. Chris Kilsby acknowledges the support of the 425 Willis Research Network. The water depth observations for Carlisle were provided by Dr. Jeffrey Neal.

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

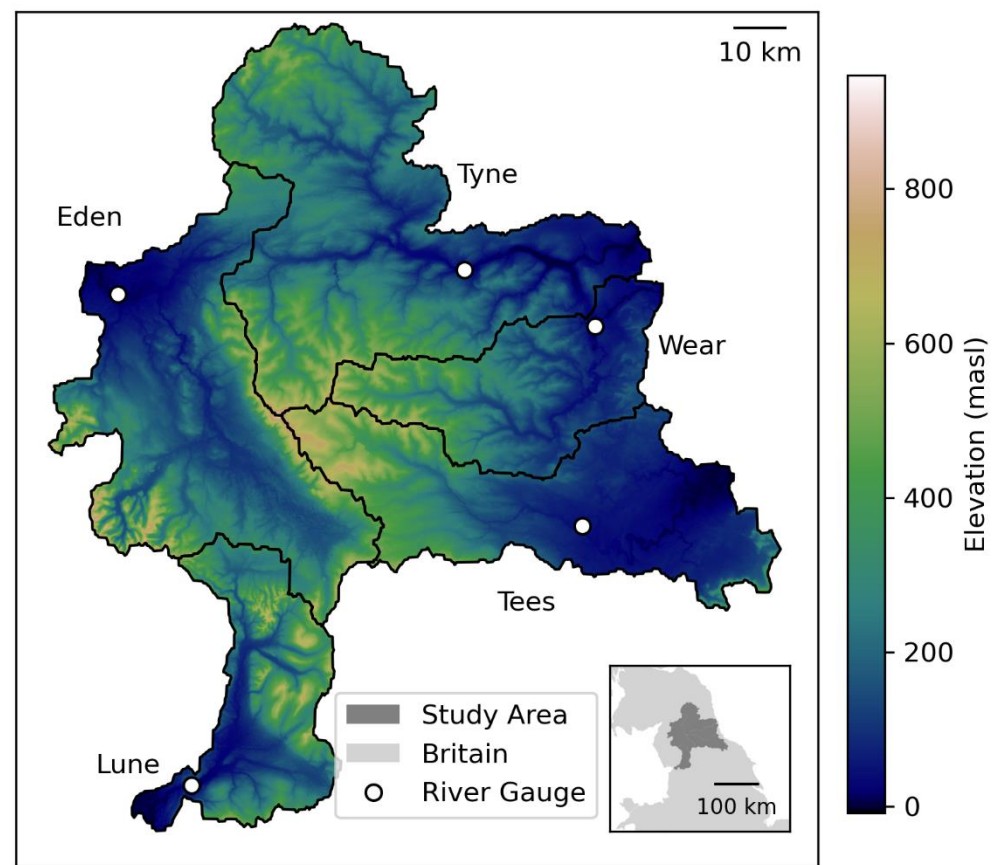

**Figure 1 The location and topography of basins within the study area, illustrated using OS Terrain 50.**

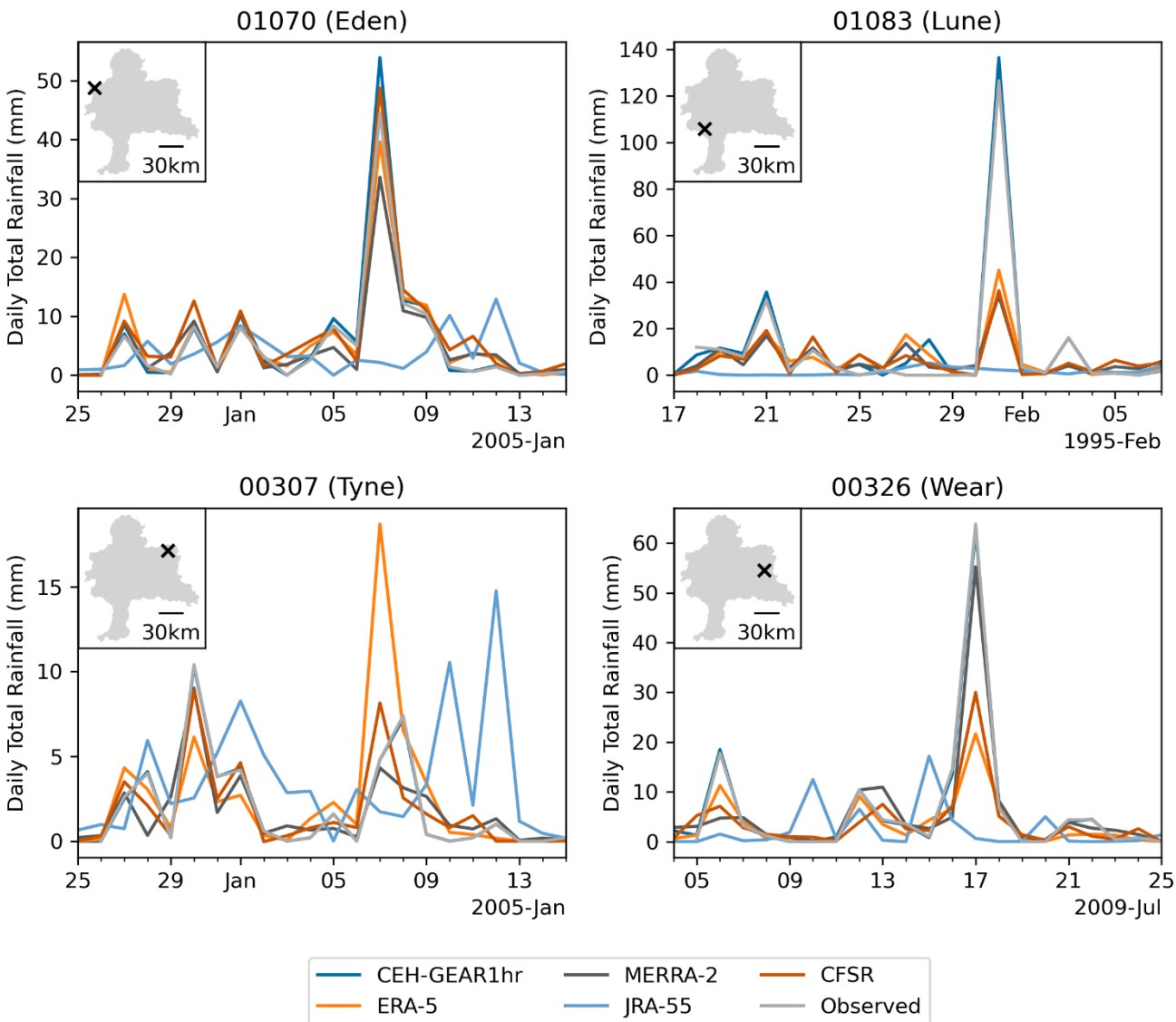

**Figure 2 – Comparison of daily rainfall totals from each dataset with observed values at selected gauges. The observations were obtained from MIDAS Open** (Met Office, 2019)**. The MIDAS station ID of each gauge is shown in the title of each subplot. The series were converted from hourly to daily to improve clarity. CEH-GEAR1hr becomes obscured in places due to it precisely following the observed series.**

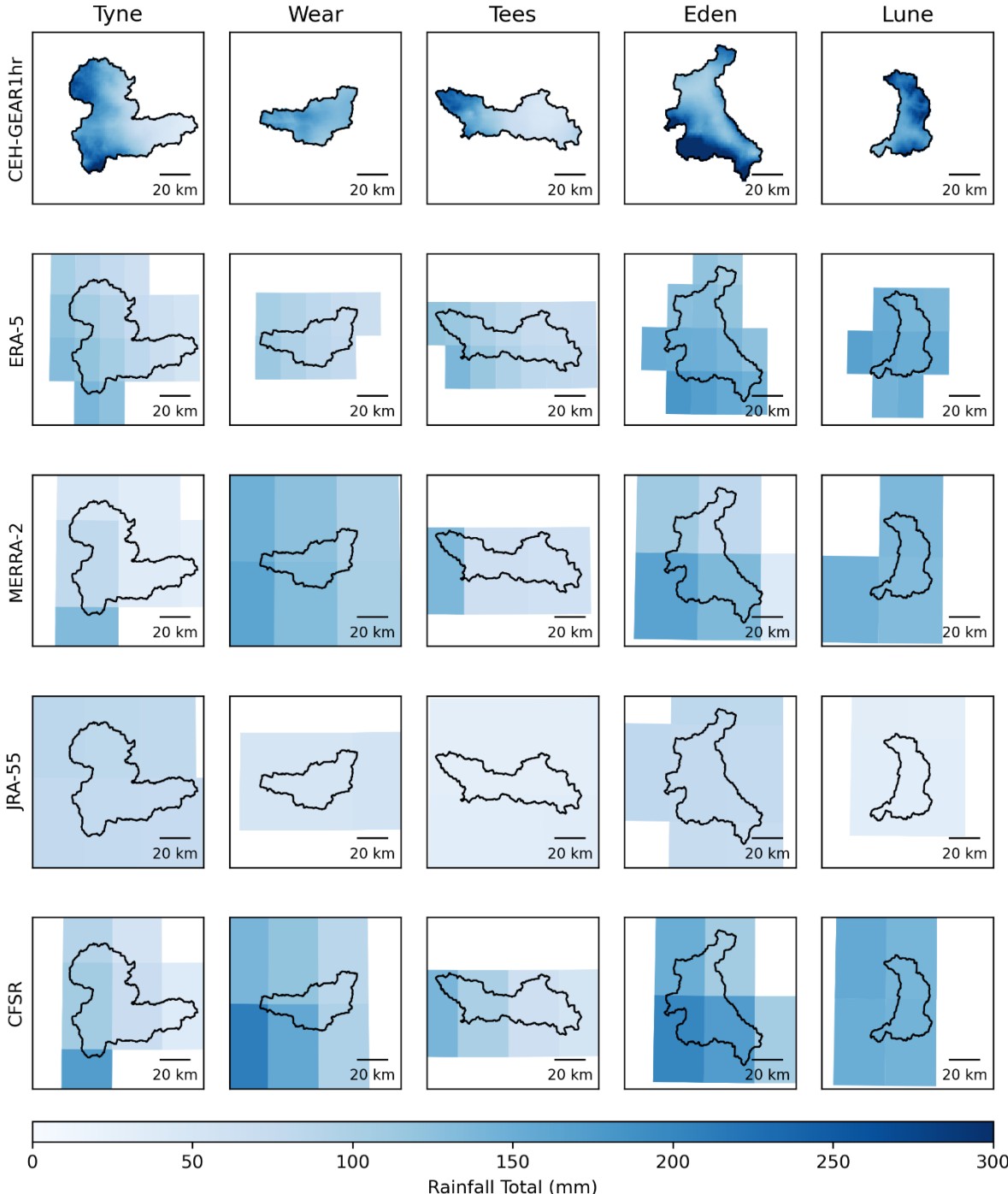

**Figure 3 Rainfall over the study area during the events in Table 2, according to each reanalysis dataset. The 1995 event is shown for the Tees and the Lune, the 2004 event is shown for the Tyne and the Eden, the 2009 event is shown for the Wear.**

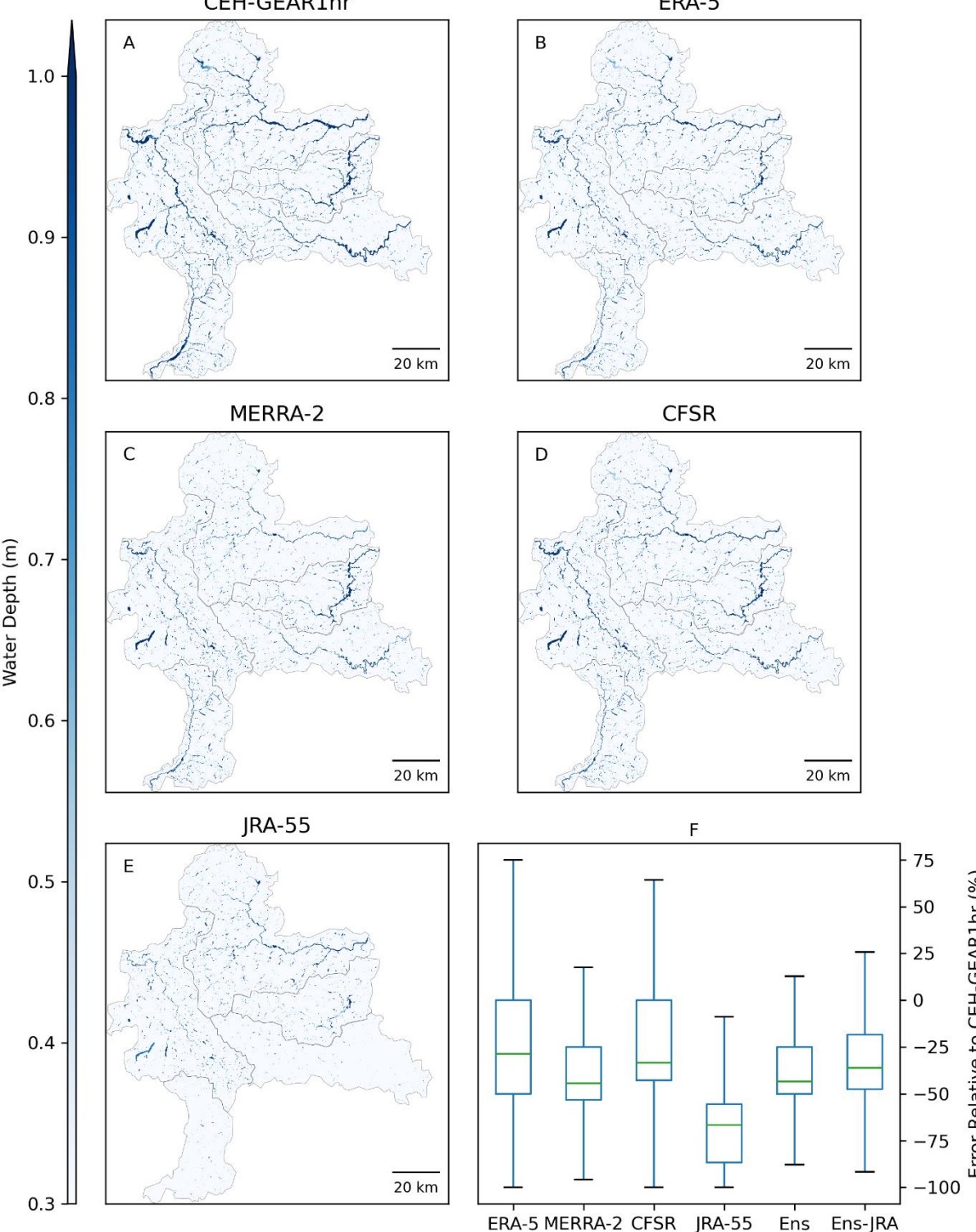

**Figure 4**

**(A)-(E) show maximum water depth throughout the study area from models using each of the rainfall datasets. (F) shows the depth error of the reanalysis datasets relative to CEH-GEAR-1hr across all cells, excluding outliers. "Ens" refers to the ensemble mean of all reanalysis products. "Ens-JRA" refers to the ensemble mean of reanalysis product excluding JRA-55.**

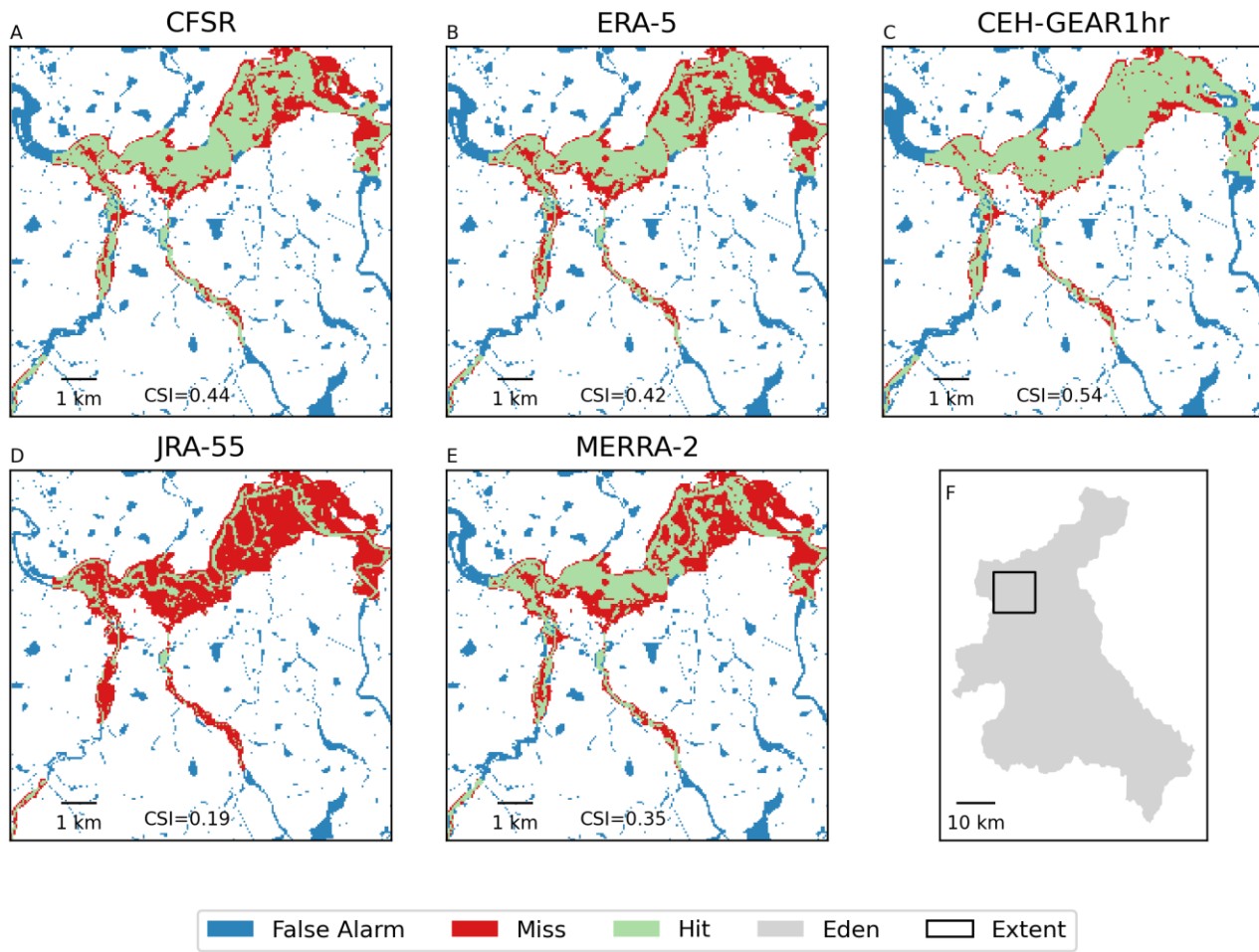

**Figure 5 Comparison of flood extent based on a threshold of 0.3m with Environment Agency Recorded Flood Outlines for the 2005 event in Carlisle.**

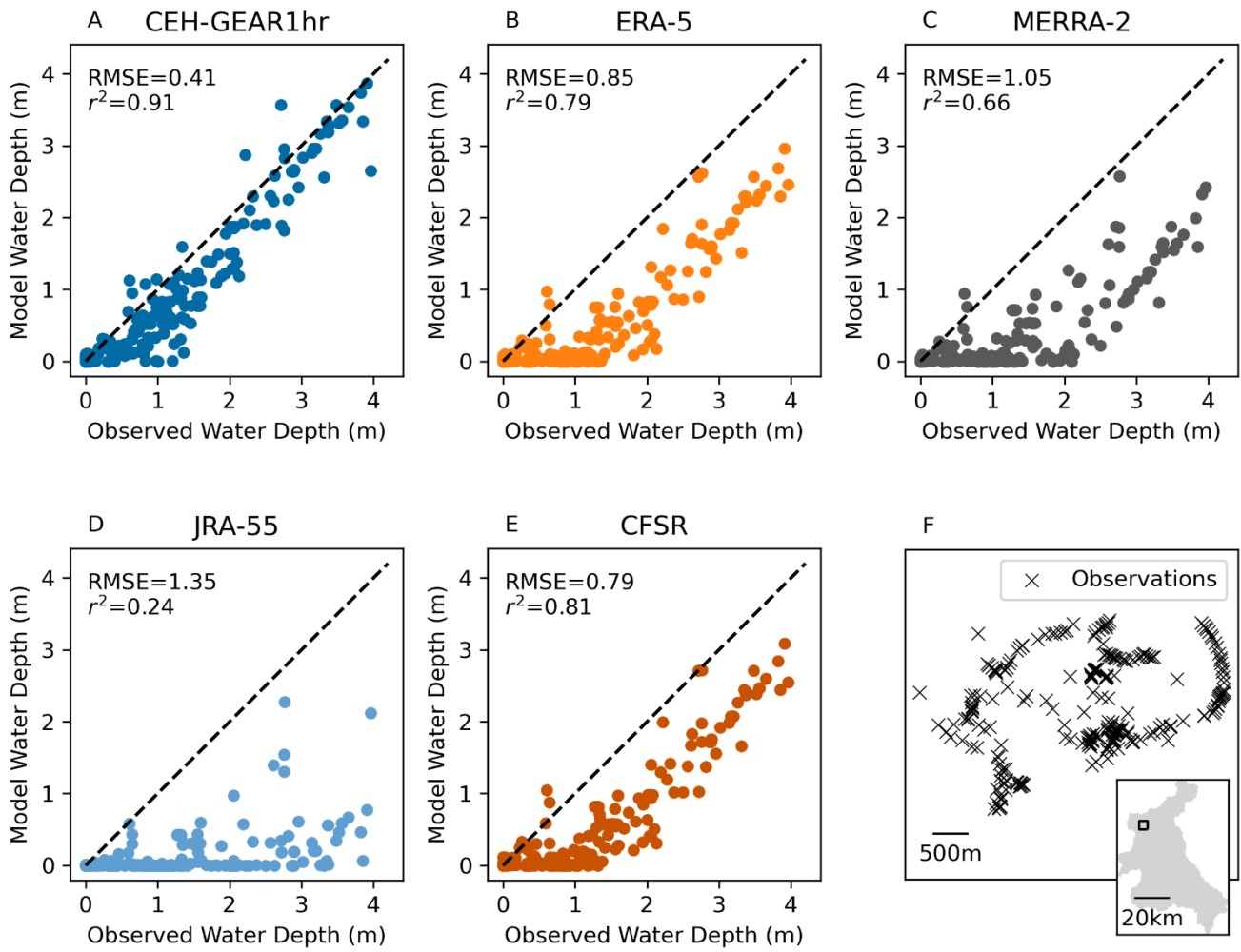

**Figure 6 (A)-(E) show a comparison of modelled and observed water depths from wrack and water marks in Carlisle during the 2005 event** (Neal et al., 2009)**. (F) shows the locations at which the water depths were measured.**

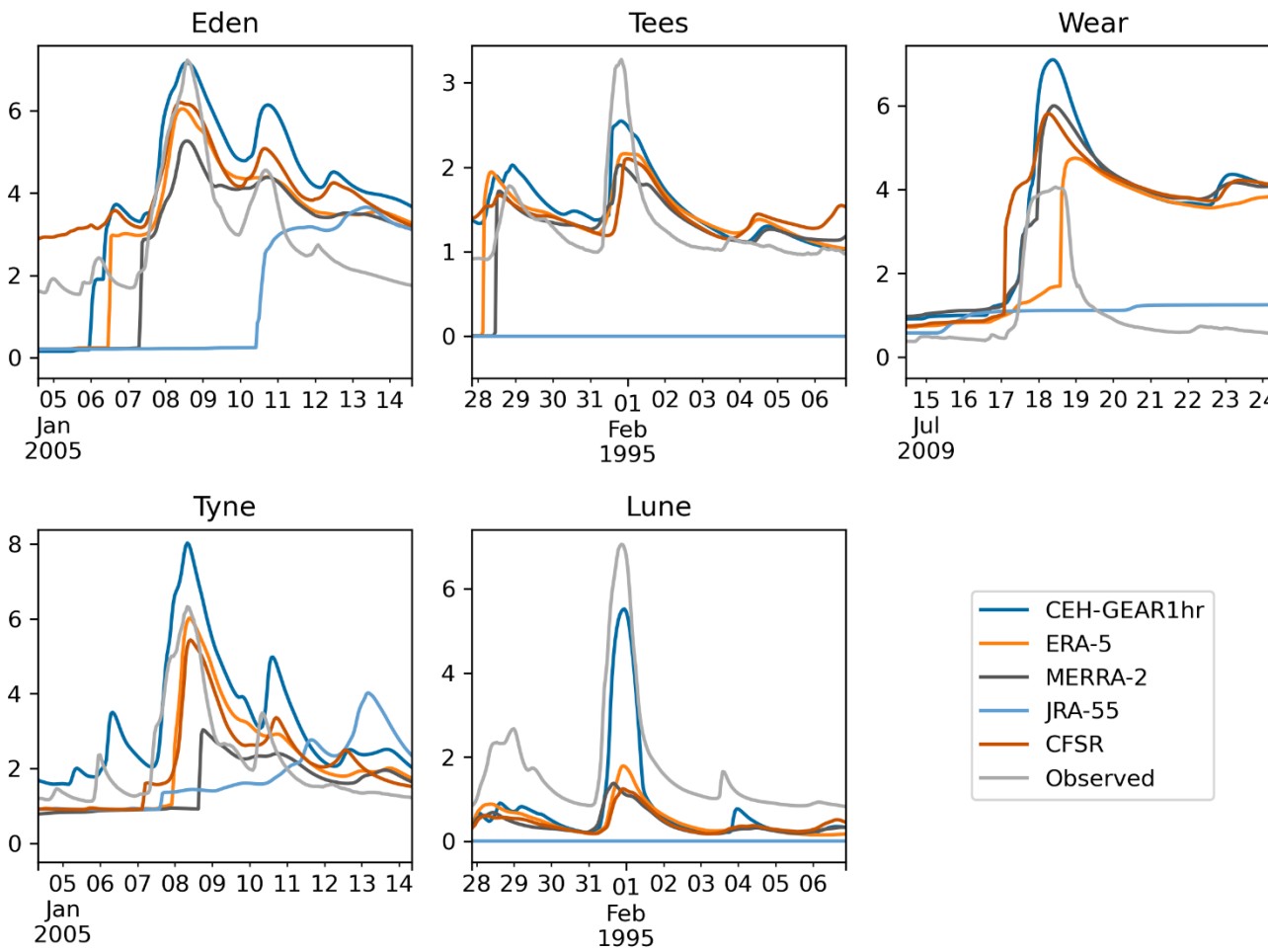

**Figure 7 Stage hydrographs comparing water depths (m) from model results and observed values at each gauge.**

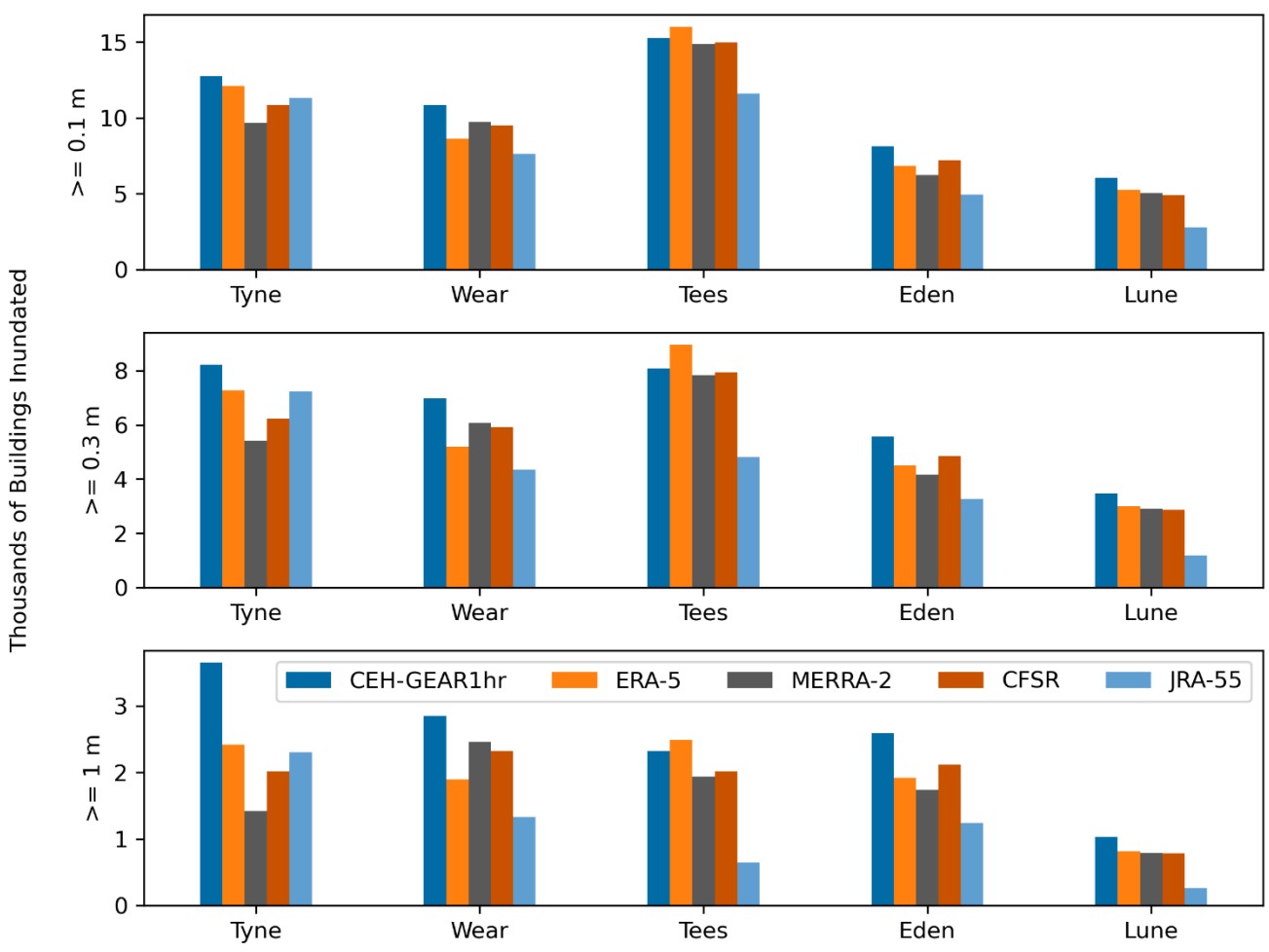

**Figure 8 Number of buildings inundated above given thresholds per basin by each model.**

**Table 1 Precipitation products included in this study. Where the end of the period is not given, the product continues to be updated to the present day at the time of writing.**

| Dataset | DOI | Resolution | Coverage | Period | Frequency |
|---|---|---|---|---|---|
| CEH-GEAR1hr | 10.5285/d4ddc781-25f3-423a-bba0-747cc82dc6fa | 1 km | Great Britain | 1990-2014 | Hourly |
| ERA-5 | 10.24381/cds.adbb2d47 | ~30 km | Global | 1979- | Hourly |
| MERRA-2 | 10.5067/7MCPBJ41Y0K6 | ~55 km | Global | 1980- | Hourly |
| CFSR | 10.5065/D6513W89 | ~35 km | Global | 1979-2011 | Hourly |
| JRA-55 | 10.5065/D6HH6H41 | ~60 km | Global | 1958- | 3 Hourly |


**Table 2 Event start and end times for each basin based on the observed stage peak at the most downstream gauge. Start times are two weeks before, and end times two weeks after, the observed stage peak times to allow for model spin-up and inclusion of**

hydrograph. Rainfall totals are the mean rainfall between the start and end times across the gauged catchment from CEH-GEAR1hr.

| Basin | Gauge Location | Gauge ID | Catchment Area (km2) | Rainfall Total (mm) | Stage Peak (m) | Flow Peak (m3/s) | Peak Time | Start Time | End Time |
|---|---|---|---|---|---|---|---|---|---|
| Wear | Chester le Street | 24009 | 1008.3 | 151.7 | 4.1 | 378 | 18/07/2009 11:00 | 04/07/2009 11:00 | 25/07/2009 11:00 |
| Tyne | Bywell | 23001 | 2175.6 | 169.2 | 6.3 | 1390 | 08/01/2005 08:00 | 25/12/2004 08:00 | 15/01/2005 08:00 |
| Tees | Darlington Broken Scar | F3501 | 818.4 | 163 | 3.3 | 646 | 31/01/1995 20:15 | 17/01/1995 20:15 | 07/02/1995 20:15 |
| Lune | Caton | 724629 | 983 | 216.1 | 7.1 | 1400 | 31/01/1995 21:15 | 17/01/1995 21:15 | 07/02/1995 21:15 |
| Eden | Sheepmount | 765512 | 2286.5 | 216.5 | 7.2 | 1520 | 08/01/2005 14:30 | 25/12/2004 14:30 | 15/01/2005 14:30 |


Table 3 Summary of metrics for each model. CSI and water depth RMSE are reported for regions of the Eden basins corresponding to those shown in Figure 4 and Figure 5.

| | | CEH-GEAR1hr | ERA-5 | MERRA-2 | JRA-55 | CFSR |
|---|---|---|---|---|---|---|
| Stage peak error (%) | Wear | 74.99 | 17.02 | 47.92 | -68.55 | 43.29 |
| | Tees | -22.28 | -34.11 | -38.16 | -100 | -35.9 |
| | Eden | -0.9 | -16.39 | -27.17 | -49.47 | -14.24 |
| | Tyne | 26.88 | -4.91 | -52.04 | -36.43 | -14.09 |
| | Lune | -21.81 | -74.66 | -80.57 | -99.99 | -82.37 |
| Stage peak time error (hrs) | Wear | -2 | 13 | -1 | 168 | -5 |
| | Tees | 0 | 2 | -1 | -336 | 4 |
| | Eden | -1 | -3 | 0 | 114 | -5 |
| | Tyne | 0 | 2 | 10 | 116 | 2 |
| | Lune | 2 | 1 | -5 | -79 | 1 |
| Number of buildings inundated >= 0.3m | Tyne | 8230 | 7287 | 5405 | 7240 | 6244 |
| | Tees | 8078 | 8956 | 7834 | 4801 | 7938 |
| | Wear | 6979 | 5187 | 6070 | 4357 | 5923 |
| | Eden | 5573 | 4515 | 4167 | 3262 | 4843 |
| | Lune | 3475 | 3005 | 2905 | 1177 | 2861 |
| CSI | | 0.54 | 0.42 | 0.35 | 0.19 | 0.44 |
| Water depth RMSE | | 0.41 | 0.85 | 1.05 | 1.35 | 0.79 |