# Peer review of "Intercomparison of global reanalysis precipitation for flood risk modelling"

_Hydrology and Earth System Sciences, 2021_

## Author Response (AR1)

We would like to thank both reviewers for their thorough scrutiny of the paper. We have addressed all their comments, which have made the paper clearer and more robust.

Two versions of the revised paper are submitted, one of these has tracked changes. In the responses to points raised by the reviewers, we provide the line numbers for where the point was addressed in the version with tracked changes.

**Response to points raised by Reviewer 1**

*RC1.1 Pg 2 "Study Area" section: manuscript would benefit from a little more information on the hydrological data used. For example, what is the source, what is the time step (i.e. 15-min aggregated to hourly?), what are the areas of gauged catchments in Figure 1?*

The source of the river level data and time step have been indicated in the text (L204). The catchment areas, locations and ID numbers of each gauge have been added to Table 2.

*RC1.2 Pg 2 and 3 "Model Setup" section: There is very little detail given on the CityCAT model and experimental set-up for the reader. Additional detail on the basic model structure, model assumptions, and how the different precipitation datasets were pre-processed to run through it would be useful. For example, how were the reanalysis grids downscaled to the respective catchments?*

Further clarification has been added about rainfall pre-processing (L129) and additional detail has been added about the hydrodynamic model and assumptions (L83-92).

*RC1.3 Pg 5 "Figure 2": is a little confusing what is shown. Is it the average of the 5 events shown in Table 2 for each of the 5 datasets? Please confirm and expand in the main text.*

Figure 2 has been updated so that each event in each basin is shown within its own subplot (25 in total). The bar plot of mean totals across events has been removed as it was causing confusion.

*RC1.4 Pg 8, Figure 5: Am I correct to assume it cannot be concluded which dataset has the most accurate number of buildings inundated? I.e. there is no true estimate from e.g. insurance claims for these 5 flood events? Should the reader assume CEH-GEAR1hr is closest to reality as by nature it is based on observations and is higher resolution?*

It has been stated in the text that there is no observed building inundation data available but CEH-GEAR1hr is likely to produce the best estimate (L263)

*RC1.5 Pg 10, L247: What is the reason for "excluding JRA-55" here?*

JRA-55 is far outside the range of results produced by the other reanalysis products and was therefore excluded as an outlier. This has been clarified in the text (L333).

*RC1.6 Pg 10, L260-262: "(...CFSR only inundated on average 14.4 % fewer buildings than CEH-GEAR1hr), caution should be used when interpreting outputs from any models based on them". I think it's difficult to jump to such a conclusion based on the fact that we do not know the underlaying CEH-GEAR1hr ability to capture building inundation across the 5 flood events in reality. Please qualify.*

Further qualification about the lack of observed building inundation data has been added in the text (L350-351).

*RC1.7 Pg 10, L265-266: "JRA-55 should not be used in flood risk modelling". This is a very strong conclusion and given your assessment is only over 5 flood events, I would argue it's too strong. Please moderate recognising the limited sample set of events used.*

This statement has been moderated and the small sample size acknowledged (L355-356)

*RC1.8 Pg 4: L126-127: The main ERA5 paper is now published by Hersbach et al. (2020) and might be useful to add*

The ERA-5 reference has been updated to Hersbach et al. (2020) (L172)

*RC1.9 Pg 6: Table 3: Missing "Building" in "Mean Absolute [Building] Inundation Error" in table column header?*

Table 3 no longer includes mean absolute inundation error and instead lists the total number of buildings inundated by each model

**Response to points raised by Reviewer 2**

*RC2.1 The key assumption underpinning the paper is that the national gauge-based rainfall product is significantly more accurate than the global reanalysis data, and results in flood simulations which can be considered as synonymous with (or at least much closer to) 'truth' than the other model realisations. If you cannot make this assumption, then the analysis is reduced in value significantly. Whilst there may be some good reasons to believe local gauge-based products are better, these are never explicitly discussed or backed up with evidence. I think there thus needs to be a robust discussion of the likely quantitative errors in CEH-GEAR1hr. A particular worry in this respect is Figure 4, which compares downstream observed river stage hydrographs to the same quantity in simulations driven by the reanalysis and benchmark precip data. However, it is not immediately clear to me from Figure 4 that the CEH-GEAR1hr simulation is significantly better than the simulations using ERA-5, MERRA-2 and CFSR. JRA-55 is clearly poor, but the other reanalysis products seem to do quite a good job given other errors in the modelling process. For the paper to be viable I would want to see more compelling evidence and arguments that the benchmark data really does provide a definitive point of reference. Figure 4 is the only absolute test of this in the paper and the results are not obviously conclusive. Properly quantifying the model performance shown in this figure with a basket of metrics including NSE and RMSE will be important and will perhaps show what I am missing just by eyeballing the plots. A wider range of other absolute measures of model performance (other gauge sites, flows as well as just stage, inundation observations if available) would also help convince the reader that the benchmark is robust.*

A discussion of potential sources of error in CEH-GEAR1hr has been added (L144-146). Flood extents have been compared to Environment Agency recorded flood outlines (L233-L241) and depths have been compared to point observations (L243-L250) from the 2005 flood event in Carlisle. These results demonstrate that CEH-GEAR1hr consistently performs better than the reanalysis products when compared with observations. Table 3 has been updated to show disaggregated metrics for each basin. This highlights the skewing effect of the Wear results which are not reliable as the gauge is inaccurate when the river is out of bank (L256). CEH-GEAR1hr had the lowest peak error in 3 out of the 4 reliable gauges. Furthermore, the errors relative to CEH-GEAR1hr have been removed from Table 3 and all values are now relative to observations.

*RC2.2 A second issue is that the analysis jumps straight to comparisons of hydrodynamic model output, and whilst this is interesting, I think the paper is missing a trick by not first simply analysing the differences between the various rainfall products. This should explain a lot about the differences in model performance that then follow. At present only Figure 2 really does this, but it is not a detailed enough dive into the differences between the precipitation data sets. As well as CEH-GEAR1hr I would also have liked to see data from the individual rain gauges across the study catchments and how the gridded products compare to these.*

Rainfall values have been compared with observations at gauges in four locations (L119-L125, Figure 2). CEH-GEAR1hr was a direct match for the observed gauge values in many places. This is because CEH-GEAR1hr was created using observations from gauges. Reanalysis datasets tended to under-estimate gauge observations. This has been identified in the text.

*RC2.3 I felt the paper was missing a lot of background information that I was expecting. Individually, each bit of missing information is minor, but taken as a whole I'm not able to really understand key aspects of how the analysis was undertaken. Just in terms of the model as one example, the paper is missing information on the numerical scheme, the grid resolution (I assume this is the same as the terrain data but you never say), how the river channels are handled (or not) and information about model boundary conditions etc. There are lots more comments like this below and they all need sorting out.*

Details have been added about the numerical scheme, grid resolution, river channels and boundary conditions (L83-92)

*R2.4 I was just a little bit underwhelmed by the volume of analysis given the amount of effort that has obviously been spent wrangling the data into shape. The model simulations are inter-compared in terms of only a handful of metrics which are either aggregated over the whole area or over all events or are for just a single location in each catchment. I felt you could have exploited the hard work you have undertaken a lot more effectively and that this would have told a richer and more interesting story. There are many more gauge sites within each catchment for example, and many of these also record flows. In particular, more absolute validation is, I think, essential to increase confidence in this study.*

Results have been disaggregated in Table 3 to demonstrate variability of metrics between basins. Additional analysis has been added looking at flood extent compared with EA recorded flood outlines (Figure 5) and wrack/water mark measurements (Figure 6) for the 2005 event in the Eden. The most downstream gauges were used in each catchment as the upstream areas are largest, and therefore more of the reanalysis rainfall data is included in the modelling (L205-206).

*RC2.5 For Figure 4, I don't see how you can predict stage accurately when you don't seem to have river channels explicitly in your model. You do not mention bathymetry or channel data, and the model grid appears to 50m resolution which will not resolve the channels. I don't think the OS Terrain 50 data you use contains the channel geometry either. Maybe I am missing something, but if you do not have the channels explicitly represented then I don't see how you are able to simulate a reasonable stage-discharge curve at the gauge sites?*

*RC2.6 On a similar note, the assumption that the subsurface hydrology is not important during these events is quite a big leap. This and point 5, would be (just about) fine if you were just doing a relative comparison, but to drive home the message in the paper you really do need to demonstrate that the benchmark is fundamentally better through an absolute validation. Given the lack of (i) channels and (ii) subsurface hydrology how does the model even get close to simulating stage correctly as shown in*

*Figure 4? I completely accept that it does, but it seems counter-intuitive. Are there some compensating errors going on perhaps?*

A justification for not including channels (L88-92) and ignoring infiltration (L95-99) has been added to the text drawing upon work by Neal et al (2021), Dey et al. (2019) and Hossain Anni (2020), and further evidence of the accuracy of simulations included relative to measurements of wrack/water marks. It is possible that some compensating errors are present with numerical dispersion and underestimation of rainfall counteracting the effect of missing infiltration, however that has not been investigated in this study.

*RC2.7 Line 28. For general readers it would be helpful to briefly explain what reanalysis products are and how they are constructed.*

A brief explanation of what reanalysis products are and how they are created has been added (L32-34)

*RC2.8 Line 28. Please define large-scale.*

Large-scale has been replace with continental- and global-scale (L34)

*RC2.9 Line 31. 'vast' is not typical scientific language. 'Extensive' would be better.*

Vast has been replaced with extensive (L36)

*RC2.10 Line 42, Define and explain VIC. General readers will not know what this is.*

VIC has been referred to more generically as a macroscale hydrological model as the specific model used is not relevant (L47)

*RC2.11 Line 51. Please state what Winsemius et al found in their study.*

The findings of Winsemius et al (2013) have been summarised (L55-56)

*RC2.12 Line 56. This is being a bit picky, but the claim here is that the results of this study are transferable to other areas bears closer examination. Is there any evidence that this is really going to be the case? The review in the paragraph above shows that reanalysis errors are complex in time and space, so this might indicate that the results of the present study are much less transferable than this statement supposes. You've only looked at five events in one particular part of the world, so it is not a very large sample.*

The limited transferability of results has been acknowledged (L64-65)

*RC2.13 Line 79. I wasn't sure what you meant by 'uniformly gridded' here. Do you mean that a regular grid geometry is used, or do you mean that each grid cell has a single uniform elevation value (i.e. what would be a p0 discretization in a finite element model)?*

The meaning has been clarified by describing the elevation surface as being made up of uniformly sized square grid cells (L82)

*RC2.14 Line 82. To what extent do rivers in HydroBASINS line up with the valley structures in the OS DEM data. HydroBASINS was generated from SRTM so my working assumption would be that there are some areas of significant mismatch between the hydrography and the terrain. This is pretty much inevitable when you mix products from different terrain data sources.*

It is true that HydroBASINS catchment boundaries may not exactly match the catchment boundaries in OS Terrain; however, they provide a reasonable estimate, and any discrepancy will not have a significant effect on results given the 50m resolution of the model grid.

*RC2.15 Line 87. There is lots of information about the model set up missing from this section. See comments above.*

As noted in the responses to earlier comments (RC1.2, RC2.3) we have provided the additional model set up information.

*RC2.16 Line 96. Is using the most extreme events the best research design? Would a mix of event types and magnitudes have been better? Extreme events tend to be valley filling which means some of your metrics may have reduced sensitivity.*

An explanation for why the most extreme events were used and an acknowledgement of the fact that looking at a range of events may have been more comprehensive have been added (L108-110).

*RC2.17 Line 101. But the land surface is assumed impermeable so how does antecedent rainfall affect the model? This statement seems at odds with the physics the model includes.*

The reason why antecedent rainfall is required has been clarified in the text (to initiate normal flow conditions in river channels) (L115-117)

*RC2.18 Table 2. Some more information on these events would be useful. Climatology, return period or rainfall and flow, dynamics etc.*

Rainfall totals from CEH-GEAR1hr over the gauged catchment areas and flow peaks have been added to Table 2.

*RC2.19 Line 112. This needs a detailed discussion of likely errors in the benchmark. You have not conclusively established that it is fit for purpose.*

A discussion of the quality control procedures applied to gauge data used in CEH-GEAR1hr and likely errors in the dataset (wind under-catch, network density) has been added (L138-147).

*RC2.20 Line 118. Some more quantitative detail about what we already know about the reanalysis errors is needed here. There is likely to be a lot of this, so it needs to be summarised effectively. So far you just have qualitative statements.*

Quantitative information about error metrics from previous studies has been added (L152-192).

*RC2.21 Figure 2. The reanalysis ensemble mean would be interesting, and the ensemble of ERA-5, MERRA-2 and CFSR.*

Figure 3 (which used to be Figure 2) has been reorganised and there was no space for additional subplots (see RC2.22), however the ensemble mean with and without JRA-55 have been added to the box plot in Figure 4

*RC2.22 Figure 2. Is this the sum rainfall from all events? Might it not be better to pick a single event as an example, and have similar plots for the other events in SI?*

Figure 3 (which used to be Figure 2) has been modified so that each event in each basin is shown within its own subplot (25 in total).

*RC2.23 Line 145. The statement that the DEM is based on airborne LiDAR cannot be true for the whole area, can it? I did not think we yet had complete LiDAR coverage of upland areas. In your previous paper you say OS terrain 50 has vertical RMSE of 4m compared to ground control points, but LiDAR data are typically accurate to <10cm. How do you reconcile this if OS Terrain 50 is LiDAR-based? Why didn't you use the available bare earth Environment Agency LiDAR where available? Lastly, how can you predict stage (cf. Figure 4) well with DEM data that have 4m vertical error?*

The source of data for OS Terrain 50 has been corrected to photogrammetry and topographic surveys (L197). An explanation for why EA LIDAR data was not used has been added and the potential for uncertainty caused by the DEM has been acknowledged (L198-202).

*RC2.24 Line 147. So, are the DEM resolution and the model resolution the same? What do you do about channels?*

The DEM and model resolution are the same and channels are not included. This has been clarified in the text (L85-86 and L88).

*RC2.25 Table 3. Some more information on the observed hydrograph data would be helpful. The circles in Figure 1 aren't really enough. What do the flow hydrographs look like at different gauge sites in the catchments and what are the event return periods?*

The ID numbers, locations, catchment areas and flow peaks of each gauge have been added to Table 2, along with the rainfall totals for the events. The timestep of the hydrograph data (15 minutes) has been noted in the text (L204). The most downstream gauges were used in each catchment as they capture a larger region of the reanalysis data (L205).

*RC2.26 Table 3. How did you calculate the inundation metrics? Particularly, how did you define the floodplain areas?*

Inundation error has been replaced by the number of buildings inundated in each model and the floodplain errors have been replaced by CSI in Table 3.

*RC2.27 Table 3. Why is the peak Q error so much bigger than the inundation error? Is this because these events are largely valley filling?*

As building inundation error is based on results from another model using the same DEM, it is expected to be lower than stage peak error, which is relative to observations. The fact that extreme events are likely to be valley-filling further contributes to a reduced variability of inundation error.

*RC2.28 Line 168. Panels a-e in Figure 3 are too small to be able to see this detail so the reader cannot verify these statements for themselves. Needs fixing.*

The size of panels A-E in Figure 4 (which used to be Figure 3) have been increased by moving the subplots onto 3 rows.

*RC2.29 Line 174. Why is out of bank flow an issue? I did not think you have channels in the model so this is a bit odd. In fact, how the model predicts stage without channels explicitly represented is surprising. See comments above on this.*

It has been clarified in the text that the out-of-bank flow is an issue for the observed values and therefore means that error is likely to be overestimated at this gauge (L256-257).

*RC2.30 Figure 3. The grid lines in panel f undermines the clarity of this diagram.*

The gridlines have been removed from the boxplot in Figure 4 (which used to be Figure 3)

*RC2.31 Figure 4. Why do some panels have a zero on the y axis and others do not?*

All panels now label zero on the y axis in Figure 7 (which used to be Figure 4)

*RC2.32 Figure 5. These numbers are not that different apart from JRA-55. Is this because the events are valley filling? In which case number of buildings inundated may not be a great choice of metric. Loss might have been a better one as that has a depth dependency.*

The reanalysis products (excluding JRA55) are 14-18% lower than CEH-GEAR1hr on average (above 0.3m), which is seen as being significant. Results comparing inundation across other inundation thresholds (0.1m and 1m) have been added to Figure 8 (which used to be Figure 5).

*RC2.33 Line 192. It would be helpful to explain this underestimation bias in physical terms.*

A physical explanation of the underestimation bias has been added (L277-280)

*RC2.34 Discussion and conclusions. These sections will need careful editing once my comments have been dealt with as this might change the inferences that can be drawn from the work.*

The discussion and conclusions have been updated to reflect the addition of comparisons with recorded flood outlines and wrack/water mark depths.

*RC2.35 Line 142. You do not compare to river flow in this paper, so this statement seems out of place.*

The reference to river flow has been changed to river stage (L327).

---

## Author Response (AR2)

Dear Editor,

We thank you and both reviewers for your time and effort to review and handle this paper. We apologise for delays in our response caused by a combination of personal, work and health factors, and thank you for allowing us to resubmit following an extension.

We thank R#1 for approving our previous edits and clarifying that they want no further changes.

We are especially appreciative of the generous time given by R#2 who has read the paper closely and identified some important points for clarification and correction.

We explain how we have addressed their comments point by point below and feel the paper is consequently far more robust. We have also added further clarification and emphasised the key point of this work which is focused on global rainfall product intercomparison.

Kind regards,

Fergus McClean and co-authors

**1. Lack of river channels in the hydrodynamic model. The revised paper argues that representation of river channels is not necessary for large events and cites two papers that it is claimed back this up (new text on lines 88-93). I'm a co-author on one (Neal et al, 2021), and I have to say this is absolutely not what our paper shows. In Neal et al (2021) there is no comparison of models with and without bathymetry, but there is in his Jeff's 2012 paper:**

Neal, J., Schumann, G.J.-P. and Bates, P.D. (2012). A simple model for simulating river hydraulics and floodplain inundation over large and data sparse areas. Water Resources Research, 48, Paper no. W11506, (10.1029/2012WR012514

And what this shows is crystal clear: representation of channels is critical for correct simulation of flood propagation. In fact, the reason for developing the different ways of representing channels in Jeff's 2021 paper was precisely because the 2012 findings showed that this was so necessary. The paper by Dey et al (2019) has also been misunderstood. Like Neal et al (2021), Dey et al compare different ways of implementing the bathymetry in large scale hydrodynamic models and also do not ever consider the 'no bathymetry' case. The statement quoted in the text that "the choice of bathymetric model becomes irrelevant at high flows for predicting hydraulic outputs" does NOT mean that bathymetry irrelevant at high flow as claimed. Rather, it says that the choice of channel shape and how much longitudinal spatial variability needs to be included is unimportant, but the paper is clear that a channel of some form is still needed. In fact, two sentences on from the quoted test, Dey et al make the statement "1D hydraulic models for high flows do not require incorporation of geometric variability in channels and even using a simple triangular or rectangular shape is sufficient for flood modeling purposes". Dey et al's opening sentence is "Accurate representation of river bed topography, commonly referred to as bathymetry, plays a critical role in a variety of hydrologic and hydraulic applications including but not limited to flood modeling". The papers by Dey et al and Neal et al are remarkably consistent, but do not show what you claim. Moreover, the importance of bathymetry has been recognised all the way back to the very earliest papers on large scale hydraulic modelling e.g. Kate Bradbrook's work for JBA in the early 2000s.

The use of citations in the revised paper to support not including bathymetry is therefore both wrong and selective: you must surely have read the rest of the paragraph in Dey et al to see that they flatly contradicted the point you were seeking to make only a couple of sentences further on?

Moreover, there are good physical reasons why channels are central to flood routing and inundation prediction even during extreme flows. First, even at high flows a good proportion of the discharge still goes through the channel because of the high velocities there. From flume experiments and field observations we know that floodplain flow velocities are often an order of magnitude lower than those in the channel, so channel conveyance is still a major component of the total flux. Let's just think about the Carlisle 2005 event. In Carlisle, the Eden is ~70m wide and ~5m deep. With a conservative average channel flow velocity during the event of 1m/s that's 350 m3s-1 out of a total flow of around 1600 m3s-1. And in terms of channel flow:floodplain flow ratio, Carlisle 2005 is likely to be an extreme (low) end member. Channel velocities during the 2005 flood were probably even higher, and the event itself was 150 year return period so there was a lot of floodplain inundation. If you have no channel, then the unaccounted for channel flux has to be distributed over the floodplain and that must lead to a significant overestimation of flows there. Second, flood waves do not propagate in a physically realistic way without a fast-moving channel flow filament set within a slower moving floodplain background field. The Neal et al (2012) paper shows this very clearly, even when momentum exchange between channel and floodplain is not included.

I fully accept that your model is getting reasonable results, but the justification for not including bathymetry does not stand up to scrutiny, and actually runs counter to everything we know about channel-floodplain hydraulics. The errors so generated will be significant (my back of an envelope calculation using the real geometry suggests ~20% of the discharge for Carlisle 2005) and leads to the suspicion that your model may only be getting the right results for the wrong reasons.

I'm sorry, but you are going to have to provide improved text to justify not including channels. Iif plausible arguments cannot be found then you will have to be clear about the errors this can generates and explain why your model still appears to perform well in spite of these (i.e., explore what might be going on to compensate).

Thanks for this very important clarification and identifying our error. We apologise and have amended the paper to correctly represent these publications.

We have added further explanation about channel bathymetry including discussion about uncertainties. We have also referenced the Neal et al. (2012) paper you cite.

We have addressed the above point(s) in several sections throughout the paper. To summarise the key points we have included are:

- No editing of the DEM data has been undertaken as the focus of this work is on the sensitivity of model performance to different global rainfall products and we have, by choice, not adjusted any other data to focus on the rainfall.
- We note that a number of studies highlight the importance of the bathymetry is reduced at higher flows, we now clearly state our assumptions will likely lead to an overestimate of discharge.

- At lower elevations, and within valleys a river channel is captured directly within the DEM. For example, as in the screenshot below the shape of the channel is rectangular. Channel depth varies from 1.5-4m, the river width ranges from 50-100m (i.e. one to two DEM grid cells).
- OS Terrain 50 is well conditioned for slope, and some key features, and whilst absolute accuracy is low, its precision (i.e. relative accuracy over a small area) is therefore much higher.
- Further discussion is added about the implications of these assumptions and around uncertainties more generally, including on the accuracy of rainfall data, drawing from some of the literature cited by the reviewer and elsewhere.

We show below a screenshot of the DEM, with the scale limited to show elevations between 4m (darker) - 30m (lighter) for the Carlisle case study site. The river channel can be clearly seen as embedded in the DEM. Whilst we hope this is useful context to show in our response, we do not believe a plot like this is required in the paper but would be happy to produce something if the reviewer and editors think this would be helpful.

[Figure]

I think you also have to better explain how the model can get the stage-discharge relationship correct, as Figure 7 seems to show, when you do not have any channel bathymetry in the model. My understanding of the model simulations is that you run a 'warm up' period to get 'normal flow' in the river channels (line 116) which I assume means below bankfull discharge for the real geometry. The event simulation then takes the flow from what would, in real life, be in- to out-of-bank flow. Without a channel geometry (however approximated) how can stage in the model respond correctly to changing discharge? Figure 7 shows that the change of stage with discharge in the model is similar to the observations, but the latter implicitly includes channel bathymetry whereas the model does not. I am at a loss to explain this and just don't understand how this can work.

We have added explanation to define normal flow.

As noted in response to point 1, we have clarified details about the river channel.  However, Figure 7 plots a timeseries of stage, we do not plot stage-discharge, which hopefully allays the reviewer's concerns as to what Figure 7 is showing.

**2. Calibration. One thing that might be a compensating effect which you don't discuss at all in the paper is calibration. How was this done, if at all, and what model parameters values did you end up using? I should probably have identified this in my previous review, but could you also add a short section outlining how the model was calibrated and commenting on the extent to which the optimum parameters are in the right physical range?**

Our focus here is on the performance of different types of globally available rainfall data.  There is therefore no specific calibration as this would compensate, differently for each catchment being studied, for the errors and differences in the data that this work is seeking to understand.  Further discussion has been added to Section 4.

The only hydrodynamic modelling parameter that has been set is Manning's *n* which was uniformly defined as 0.03 based on Chow (1959), and is consistent with many other studies.  We have now clarified this in the text and cited some other studies that justify this selection.

**3. Lack of infiltration. I was more convinced by your justification for not including infiltration during extreme events as that seems at least physically plausible, but the paper you cite to justify this (Hossain Anni, 2020) is a small-scale study in an urban catchment, so again does not show quite you claim. Please could you replace with references showing the same effect at catchment scales.**

We agree that this paper alone is insufficient on its own.  We have therefore included further evidence from other more directly comparable studies to support this statement and understand the implications of this assumption (e.g. Ni et al. 2020; Hou et al. 2021).  These report that peak flow and/or extent are insensitive to infiltration rates, but has an impact on outflows as the flood wave falls.  The effect of this is greater for longer floods, and would be more significant in semi-arid or arid regions.  Explanation has been added to Section 2.2 and in discussion in Section 4.

In the case of the Carlisle 2005 event it is widely reported that the ground was saturated, we have added a citation to this effect.

**4. Lack of flood defences. I was surprised how you got such good results in Carlisle without including flood defences. Even in 2005 Carlisle was well defended and I don't think these were always overtopped, so this needs a couple of sentences of explanation in the paper.**

Figure 4 compares observed and modelled depths, whilst Figure 5 shows how the models perform against the observed flood extent from the detailed mapping of the 2005 Carlisle event that was undertaken by: Neal, et al. (2009) Distributed whole city water level measurements from the Carlisle

2005 urban flood event and comparison with hydraulic model simulations, J. Hydrol., 368, 42–55, https://doi.org/10.1016/j.jhydrol.2009.01.026.

We have been unable to find more accurate information than that reported in the Neal et al. paper. Further, we have not been able to locate the historical defence levels for Carlisle in 2005, nor find studies that report which defences were overtopped and which were not.

Without further data it is not possible to answer this question with certainty, however, we would suggest that if there were lengths of defence that did not overtop in 2005 then those are located in parts of our model where the floodplain elevation is higher or rapidly rises, for example in the NW of the study area; or areas are flooding as floodplain flows allow water to go round the back of higher flood defences.  Consideration of the latter mechanism is not something we specifically studied in this rainfall intercomparison, and would probably benefit from a different methodology if it were to be analysed.

This has been clarified in the paragraph on the results from the Carlisle flood in Section 3.

**5. Line 116. Could you define 'normal' flow? I assume it is somewhere below bankfull discharge (see above) but I think this needs to be clear.**

Thank you, we have added an explanation to clarify what is meant by normal flow: "*Antecedent rainfall is necessary to initiate normal flow in the river channels, which requires the water from all upstream cells to reach the outlet of the basin. Normal flow here refers to the flow in the channel before the flood event took place. If no spin-up period is included, then flood magnitudes would be underestimated, and the flood wave would not propagate in a physically realistic way.*"

**6. Digital Elevation Model. I still can't quite understand what is happening with the DEM data. Thanks for confirming that OS Terrain 50 is photogrammetry and not LiDAR, but that does mean you are using a DEM with 4m RMSE of vertical error. If that is genuinely the case, how did you get inundation and water depth predicted as well as you do? Either the actual vertical error over the areas you simulate is substantially lower than 4m or there is something odd going on in the model to compensate.**

First, I think you need to be clear about the DEM error in the current paper. You don't mention this anywhere, but it is huge (DEM RMSE >50% of the observed flood wave amplitude for all catchments).

Second, you need to explain how the model is still able to produce the results it does despite these terrain data errors.

The RMSE for OS Terrain 50 is reported to be 4m across the whole country, being calculated as the absolute accuracy relative to a number of GNSS points across Great Britain.  This combines systematic (e.g. block linkages between photogrammetric observations) and random errors from one end of the country to another and is therefore not an absolute measure of accuracy for a given area of interest.

Local precision is more important here, this is the relative accuracy from point to point.  This will generally be much better than RMSE, especially over the relatively small area visualised in Figure 15.  OS

don't publish the local precision of OS Terrain 50 but the Royal Institute of Chartered Surveyors suggest (Table 5, P24) that for an Area Of Interest it would likely be of the order of decimetres: www.rics.org/globalassets/rics-website/media/upholding-professional-standards/sector-standards/land/earth-observation-and-aerial-surveys-global-guidance-note-6th-edn.pdf

Although they don't report their local precision, OS Terrain 50 has been validated to meet the positional requirements for key features such as waterbodies, and to capture slope: (www.ordnancesurvey.co.uk/documents/product-support/support/os-terrain-50-overview-v1.5.pdf ).

The combination of relative accuracy and positional validation against key features explains why the DEM is able to capture the relative positioning of key features that dominate the flood extent of major storms in this catchment. Further discussion on this and other uncertainties raised by the reviewer has been added into Section 2.4 and 4.

Lastly, your statement regarding OS Terrain 50 that it "has been shown to perform best for flood risk modelling in a comparison with other DEMs (McClean et al., 2020)" derives from a paper that did not include LiDAR data in the comparison. Instead, your previous work only compares OS Terrain 50 to a basket of global DEMs, which unsurprisingly have higher error. Numerous previous studies, starting with Sanders (2007):

Sanders BF. 2007. Evaluation of on-line DEMs for flood inundation modeling. Adv.Water Resour. 30(8):1831–43.

Have shown airborne LiDAR data sets to have considerable advantages for flood inundation modelling. In the above text from the paper you have used the word 'best' when you mean 'better' and have not qualified the subset of DEMs that your study refers to. Because of this your statement is misleading.

We fully agree, and have amended the text to use "better (relative to global DEMs)" rather than "best".

The Yunus study you quote is also not the greatest evidence here because: (a) it is a bathtub model which ignores hydraulic connectivity and (b) the events simulated are extreme and valley filling such that inundation extent becomes a very insensitive metric for comparing DEMs. There are now many papers on DEMs in flood inundation modelling and you need to consult a range of these to revise this text instead of using a self-citation in an inappropriate way.

As an aside, I still don't understand why you are not using LiDAR data where it exists? Most river floodplains in the region are now mapped and you could use these data and patch with OS Terrain 50 in hillslope areas away from the floodplain. LiDAR vertical errors are ~10cm RMSE and it is the gold standard.

We agree that high resolution LiDAR is the gold standard. The focus of the work overall has been to understand the impact of uncertainties on open and widely available data. The particular emphasis of this paper is on globally available rainfall products. OS Terrain 50 provides a convenient DEM product for this purpose because it limits additional processing as it does not require coarsening of high resolution LiDAR and/or merging of different DEM datasets which can introduce different uncertainties;

and a 50m resolution is sufficient to characterise flood extent whilst also limiting the computational expense for 2D physically-based hydrodynamic simulations.

We have added more explanation as to why we have used OS Terrain instead of LIDAR, including a reference to Sanders (2007) and other literature.

**7. Figures. Some of these are difficult to read because the panels are small compared to the detail you are trying to show or because you have used a low-resolution raster file format instead of vector. The latter issue means that text, axes and other figure elements are quite pixelated. You need to redo these images at higher resolution or (better) as vector graphics.**

Thank you for highlighting this, the resolution of all figures has been increased to 300 dpi and additionally provided separately as vector graphics.